# Women-Reported Barriers and Facilitators of Continued Engagement with Medications for Opioid Use Disorder

**DOI:** 10.3390/ijerph19159346

**Published:** 2022-07-30

**Authors:** Alice Fiddian-Green, Aline Gubrium, Calla Harrington, Elizabeth A. Evans

**Affiliations:** 1School of Nursing and Health Professions, University of San Francisco, San Francisco, CA 94117, USA; 2Department of Health Promotion and Policy, School of Public Health and Health Sciences, University of Massachusetts Amherst, Amherst, MA 01003, USA; agubrium@schoolph.umass.edu (A.G.); callaharring@umass.edu (C.H.); eaevans@umass.edu (E.A.E.)

**Keywords:** women and opioid use disorder, qualitative methods, medications for opioid use disorder, substance use treatment, stigma and substance use

## Abstract

Opioid-related fatalities increased exponentially during the COVID-19 pandemic and show little sign of abating. Despite decades of scientific evidence that sustained engagement with medications for opioid use disorders (MOUD) yields positive psychosocial outcomes, less than 30% of people with OUD engage in MOUD. Treatment rates are lowest for women. The aim of this project was to identify women-specific barriers and facilitators to treatment engagement, drawing from the lived experience of women in treatment. Data are provided from a parent study that used a community-partnered participatory research approach to adapt an evidence-based digital storytelling intervention for supporting continued MOUD treatment engagement. The parent study collected qualitative data between August and December 2018 from 20 women in Western Massachusetts who had received MOUD for at least 90 days. Using constructivist grounded theory, we identified major themes and selected illustrative quotations. Key barriers identified in this project include: (1) MOUD-specific discrimination encountered via social media, and in workplace and treatment/recovery settings; and (2) fear, perceptions, and experiences with MOUD, including mental health medication synergies, internalization of MOUD-related stigma, expectations of treatment duration, and opioid-specific mistrust of providers. Women identified two key facilitators to MOUD engagement: (1) feeling “safe” within treatment settings and (2) online communities as a source of positive reinforcement. We conclude with women-specific recommendations for research and interventions to improve MOUD engagement and provide human-centered care for this historically marginalized population.

## 1. Introduction

Opioid use in the U.S. has resulted in unparalleled rates of accidental injury, infectious disease (e.g., HIV, Hepatitis C), and premature death [1]. Opioid-related fatalities increased exponentially during the COVID-19 pandemic and show little sign of abating [2]. Some factors influencing opioid fatalities include social isolation; increased mental health issues; reduced access to treatment and services; and stress related to economic, social, and other factors. Prior to COVID-19, concerns specific to women were increasing rates of heroin use, slower decreases in rates of prescription opioid misuse [3], and increasing deaths associated with synthetic opioids and heroin [4]. Women with OUD currently remain a growing population of concern [5,6,7]. 

Sustained engagement with medications for opioid use disorder (MOUD) (e.g., buprenorphine, methadone, naltrexone) results in reduced mortality [8], lowered opioid use [9], fewer infectious disease risks [10], reduced engagement with the criminal justice system, and other positive outcomes [11]. The COVID-19 pandemic prompted innovations to reduce treatment barriers. For MOUD, this included relaxing regulations regarding take-home doses of methadone, expanding access to telemedicine, and allowing buprenorphine to be prescribed via telemedicine appointments [12,13]. These innovations show great promise for expanded access to treatment for people with OUD. However, we know the vast majority of people with OUD still do not access treatment [1]. According to 2019 estimates, less than 30% of individuals needing OUD treatment received MOUD [14]. Of those who do enter care, few remain engaged with MOUD long enough to achieve lasting recovery [15]; treatment rates are lower for women versus men [16,17]. Closing this treatment gap is paramount.

Factors that contribute to low rates of treatment engagement among women with OUD include high rates of trauma and sexual exploitation [8], mental health comorbidities [18,19], chronic pain [8,19,20], socioeconomic vulnerability, and housing insecurity [21,22]. Furthermore, women with substance use disorders face harsh stigma related to social expectations around maternal health and caregiving roles [23,24,25]; fears related to loss of child custody are substantial [26]. Finally, compared to men, women experience more heightened feelings of OUD-related shame and other internalized self-depreciating thoughts [8,27,28], resulting in fractured relationships [29] that leave women with little family and social support to remain engaged with treatment [18].

Prior research on women and substance use points to important treatment facilitators. These include trauma-informed and gender-specific treatment programs [16,26,30], positive therapeutic alliances [24], and positive social support [18,31]. Despite these insights into the women-specific causes and outcomes of substance use treatment engagement, we still know relatively little about the social and contextual factors that influence why or how women who do access MOUD remain engaged with it over time [24,32]. This article explores this knowledge gap, focusing on women-reported barriers, as well as facilitators, to MOUD engagement. Drawing from qualitative findings, we share critical context on the social and structural factors that influence engagement with the medical system among this historically marginalized population.

## 2. Methods

### 2.1. Study Design and Data Collection

Data presented are a subset from an exploratory sequential mixed methods [33] study that used a community-partnered participatory research (CPPR) approach [34] to adapt an evidence-based digital storytelling intervention [29,35] for supporting continued MOUD treatment engagement. The parent study consisted of three phases: relationship building, exploratory data collection, and patient workgroup sessions (Figure 1). Data in this manuscript draw from qualitative findings related to treatment engagement that surfaced during group and individual conversations with patients regarding adaptability and feasibility of a digital storytelling intervention during phase two of the project. Qualitative methods are best suited to generate novel findings that prompt innovations [36], of particular value when conducting research with a stigmatized population and topic (i.e., women and M/OUD).

To elicit information for intervention development, qualitative data were collected from 20 women enrolled in two outpatient Opioid Treatment Programs (OTP) located in Western Massachusetts and operated by one of the region’s largest behavioral health services providers. Both facilities are licensed to administer three FDA-approved MOUD maintenance therapies: methadone, buprenorphine, and naltrexone. Inclusion criteria included: (1) self-identified adult woman, (2) who has received MOUD from a participating OTP for at least 90 days, and (3) has no cognitive impairment that would disallow informed consent. Twenty women were recruited via flyers distributed in the OTPs, referrals from clinical staff, and participant word of mouth. Research staff conducted in-depth interviews; participants also completed brief anonymized surveys on demographics and questionnaires related to substance use history and treatment factors (Appendix A). Participants were compensated with USD25 gift cards. Interviews were digitally recorded, professionally transcribed, de-identified, and reviewed for accuracy. All procedures were approved by the OTP’s affiliated Institutional Review Board, and each participant provided written informed consent.

Data were collected from August to December 2018. In-depth group interviews were conducted with 20 patients and facilitated by the first and last author. The majority of sessions were attended by an average of 2–4 participants; two sessions were conducted with one individual each. Interview sessions lasted 1.5 to 2 h, and were conducted in private rooms at participating OTPs. Each interview opened with a grand tour prompt on barriers and facilitators to continued MOUD engagement. Remaining interview questions elicited information on attitudes, beliefs, and perceived utility of using a digital storytelling intervention to increase MOUD engagement; expected intervention outcomes; and potential challenges/solutions associated with intervention pilot-testing and further development (Appendix B). In this paper, we summarize qualitative findings from the grand tour prompt on barriers and facilitators.

### 2.2. Data Analysis

Data analysis was guided by constructivist grounded theory [37], an iterative approach that involves simultaneous data collection and analysis; inductive code development; using “constant comparison” to compare and contrast categories; memo-writing to identify and define thematic categories and any connections between them, as well as identifying gaps; and sampling for construction of meaning, not for generalizability [37,38,39]. Our analytic strategy was to examine narrative content and context [40,41,42]. Narrative content analysis focused on women-specific paradigms of OUD found in the data at both the individual and group level. Contextual analysis focused on the perceptions and structural circumstances (e.g., historical, political, economic) that shape identity and experience [41].

Informed by standard qualitative data analysis procedures [33,43,44], the first and last authors independently reviewed transcripts and conducted open-coding, using theoretical memo-writing to identify and develop evolving themes. Next, each researcher composed a list of thematic codes derived directly from the data. Then, we reviewed and compared emerging themes collectively and iteratively to reach thematic saturation and determine final themes [33,43]. During data analysis, we assessed the selected quotes to ensure they represented the diversity of participants and perspectives. Member checking during patient workgroup sessions in phase three of the project (Figure 1) further ensured trustworthiness of our thematic findings. We used reflexivity to balance interpretive authority and participants’ experiences and perceptions [33,44]; this included being conscious of power dynamics associated with conducting research with patients in clinical sites. To this last point, participants stated their eagerness to share MOUD treatment experiences with research staff that they would not share with clinical staff, expressing greater comfort in discussing these topics with individuals unaffiliated with the clinic and who do not provide care.

## 3. Results

Twenty women being treated for OUD participated in the study (Table 1). Participants self-identified as white non-Hispanic (65%), Latina (30%), and African-American (5%); the mean age was 36.6 years. Sixty-five percent reported some college or a bachelor’s degree. Only 20% of the sample were employed full-time, and 80% had an average annual income of <USD20,000. Mean duration of opioid use was 4.6 years; average duration of current MOUD treatment was 2.8 years. Below, we summarize key thematic results (Table 2), highlighting barriers and facilitators to MOUD engagement. Barriers include two themes: (1) community-level social stigma and (2) fear, perceptions, and experiences with MOUD. Facilitators to MOUD engagement include (1) a sense of safety within treatment settings and (2) social media and online communities as a source of positive social support.

### 3.1. Community-Level Social Stigma

Women in our study commonly reported feeling unable to escape the identity of “drug user loser” a stigma that “doesn’t go away until the day you die—you’re always going to be ‘that junkie.’” Despite positive treatment outcomes (e.g., abstaining from illicit substance use, employment, and maintaining child custody), women identified a persistent social discourse wherein successful engagement with MOUD did not guarantee an escape from gendered negative associations related to active substance use such as sex work and child neglect. Women reported that the social consequences of being “outed” (i.e., discovered) as a MOUD patient posed substantial barriers to continued MOUD engagement in two ways. First, MOUD directly links women to active substance use and second, MOUD is widely perceived to be an illegitimate form of treatment. Participants shared how MOUD-related community-level social stigma operates within three spaces: workplace settings, social media, and hierarchies within treatment and recovery settings.

#### 3.1.1. Workplace Settings

Workplace environments were identified as sites of discrimination, which may explain one way that being “outed” as a MOUD patient can negatively impact treatment engagement. A considerable anxiety was the potential of employers learning of participants’ MOUD status. In one group interview session, the general consensus was that women alternatively hid, or actively lied about, how MOUD treatment policies impacted job-related behaviors. For example, women agreed they would be more likely to tell an employer their child was sick, rather than explaining that lateness was treatment related (e.g., mandatory counseling or long lines). One participant who had not experienced workplace discrimination firmly believed that if she shared her MOUD status she would not be trusted by superiors or peers. She worried that if the cash register was “short” she would be blamed for stealing the money because of assumptions linking her past illicit substance use to criminal behavior.

Women in multiple interview sessions discussed their reluctance to reveal their MOUD status in the workplace due to fear of inter-employee discrimination. As one example, a woman in treatment who worked in a hospital told her peers she sometimes wants to shout “I am one of them!” when coworkers speak disparagingly about “these people, nothing but drug addicts” who enter the hospital. A second woman reported driving hours to receive treatment outside her small, rural town, because at work she hears coworkers “talking all this nasty stuff” about “drug addicts.” A third participant recounted a coworker who was open about his MOUD treatment. Although he did not experience negative repercussions related to his employment status, “everyone judged him behind his back.” As a result, that participant was adamant she would “*never*” tell anyone at work about her MOUD status.

#### 3.1.2. Social Media

In one group interview, women identified social media as a site of “constant debate”, between whether OUD is a “disease or…a choice” and a space where the validity of MOUD as a legitimate medical treatment is publicly questioned. For example, the women shared examples of memes they perceived as derogatory. The caption of one image described read: “when you’re on Suboxone you’re not really clean—it’s like the government is legal drug-dealing.” Over “400 comments” were posted in response, predominantly reinforcing this messaging. Another meme was “laughing about ‘when your girl says she’s clean’ and then it shows a picture of her walking into a methadone clinic or something,” each perpetuating notions equating MOUD to active substance use. Other examples cited were memes or posts such as “people have to pay for Epi pens but Narcan^®^ is free?” and “a person chooses to [mess] up their life and they get to get saved?”

#### 3.1.3. Hierarchies within Treatment and Recovery Settings

Across all interview sessions women shared experiences of between-women social hierarchies within treatment settings. These hierarchies position women against each other, resulting in discrimination levied from “drug addict to drug addict.” This internalized stigma manifests as a hierarchical value system within treatment and recovery spaces, which can function as an impediment to peer support and deter women from treatment engagement. Women discussed encountering such recovery hierarchies within Alcoholics Anonymous (AA) and Narcotics Anonymous (NA) chapters, some of which did not consider MOUD a legitimate component of recovery. This experience was reportedly pronounced for pregnant women and mothers, especially those living in small, rural communities. In one example, a participant—a mother—attended AA meetings in her town, but introduced herself as an alcoholic to avoid judgement and receive the peer support she identified as crucial to her MOUD engagement.

Although not self-identified as such, women appeared to internalize broader social stigma regarding women with OUD. Between women with OUD, recovery hierarchies were further entrenched by a predominating social stigma that associates women with OUD to sex work and incarceration. According to participants in a group interview session, when people “hear you’re an addict, especially heroin or cocaine, they’re like ‘…she’s a prostitute and she’s dirty.’” Participants acknowledged that fear of these associations can minimize transparency about substance use history. At the same time, participants—who were in treatment—appeared to internalize this messaging and turn that same judgement towards women actively using. For example, during that group interview participants spoke disparagingly about women engaging in sex work in exchange for drugs, referring to them as “nasty.” In that same discussion, participants in the group expressed strong opposition to providing MOUD to incarcerated individuals referring to such programs as “outrageous” and incarcerated people as not “deserving” of MOUD. Participant critiques of such programs—“you’re in jail, why are our tax dollars going to that?!” and “there should be no drugs in jail”—equate MOUD to active substance use, a contradiction that appeared to go unnoticed by participants.

### 3.2. Fears, Perceptions, and Experiences with MOUD Pharmacotherapies

This project elicited three main concerns with regard to MOUD: (1) fear of side effects and medication synergies, (2) unrealistic expectations of treatment duration, and (3) opioid-specific provider mistrust.

#### 3.2.1. Fear of Side Effects and Medication Synergies

Women identified various medication side effects and synergies, perceived or otherwise, which may negatively impact MOUD engagement. All women interviewed in the project were being treated with methadone. Two sisters who participated in one interview session reported their “ex-stepmother” was a registered nurse who believed methadone is “liquid fire that kills you from the inside out.” When asked to identify side effects of methadone complaints expressed by three different women were that “it eats at your bones,” “your teeth are all going to fall out,” and it makes women “fat.” A participant with extensive tooth decay worried aloud that “if this is happening to my teeth, I can’t imagine what’s going on with my bones.” The other women in her group interview nodded in agreement, adding that MOUD side-effects related to physical attributes such as weight gain and visible tooth decay contributed to low self-esteem. Lastly, one woman receiving pharmaceutical treatment for both OUD and mental health comorbidities raised concerns related to perceived synergies between mental health pharmacotherapies and MOUD. Women in her group agreed that common concerns regarding medication synergies included “nodding off” (i.e., increased fatigue) and dampening of MOUD efficacy.

#### 3.2.2. Expectations of Treatment Duration

When discussing patient–provider communication and education during one group interview session, participants identified frustrations related to unrealistic expectations of treatment duration. Upon enrollment, patients are reportedly told: “we’re going to get you on a steady dose and wean you off within six months.” Yet the average treatment duration among participants was nearly three years. Further, the prospect of long-term engagement with MOUD was noted to be a source of considerable discouragement. As one participant illustrated: “Every story ends up being somebody getting off [MOUD] and being great—for a very short period of time. I’ve never heard of it, actually. Of someone coming off of [MOUD] ever. I’ve never.”

Compounding unrealistic expectations of treatment duration was a purported lack of information for women to make an informed choice between MOUD pharmacotherapies (e.g., methadone, buprenorphine, or naltrexone). In a group interview discussion about medication, women interested in switching to buprenorphine expressed frustrations about feeling “stuck” on methadone treatment, expressing irritation with reported difficulties in tapering methadone doses. For one participant “Honestly, if I had the money, I would do pills for three months and then go to detox for pills…It’s five times worse to come off methadone.” Other participants in the group felt vexed that it can take “thirty to ninety days to withdraw from methadone…That is *months* of being sick. That is insane. If I had known that, I would never have gotten on it. Ever.”

#### 3.2.3. Opioid-Specific Provider Mistrust

A barrier to MOUD engagement identified in this project was increased mistrust of providers due to the iatrogenic impact of provider prescribing practices on the opioid crisis. In one group interview participant comments centering blame on physicians, for example, “I would say at least 50% of us, it (prescription opiate) was given to us by a doctor and that’s how it started,” were common. One woman additionally expressed a mistrust of providers related to conspiracy theories centered on unethical relationships between providers and the pharmaceutical industry. In agreeing with her, another woman in the group stated she was hesitant to believe that MOUD was the best treatment option “because on some end [the doctor] is also a pusher, a dealer. I’m coming here to get my fix and they’re going to benefit [by getting] X amount of dollars for each person that comes here.”

### 3.3. Facilitators to Treatment Engagement

In addition to the above findings on barriers to MOUD treatment engagement, we additionally report on women-identified facilitators of MOUD treatment engagement. Facilitators identified included feeling a sense of safety within treatment settings, and online communities as a source of support and encouragement.

#### 3.3.1. Sense of Safety within Treatment Settings

Feeling safe within treatment settings was cited as paramount for participants to remain engaged with MOUD treatment protocols, especially for those with self-identified trauma histories. Simple acts of kindness from clinicians and staff fostered a sense of safety and loyalty to a particular treatment setting. For this highly stigmatized population, having someone remember their name or share a smile left participants feeling “shocked” yet valued. Examples of positive provider encouragement included “doctors where as soon as they found out I was on [MOUD], they give me high fives and they’re like, that’s amazing!” and a clinician who told one participant “I’ve learn[ed] a lot of things from you...I learn from you, you learn from me. We learn from each other.” 

The integration of peer workers (i.e., “recovery coaches”) into the treatment setting was identified as critical for fostering a sense of safety. Peer workers were described as being an important source of empathetic and relatable support grounded in shared lived experiences. During one group interview, women recounted feeling “frustrated” when assigned to counselors who compared heroin to “sugar or soda” or suggested going “for a jog around the block” as an alternative to “using.” During this session women resoundingly reported wanting to interact with a peer worker—someone that “’shares the struggle,’ because [i]t’s not as easy as people think it is.” On-site peer workers reportedly represent a form of “hope,” in part due to their sustained recovery that is a requisite of the job. Women reported aspiring towards becoming certified as peer workers as a meaningful way to “give back” to others once they achieved treatment stability.

#### 3.3.2. Support from Online Communities

Despite being a reported source of discrimination, participants simultaneously reported positive social connection and support from online communities of people enrolled in MOUD treatment. In one group interview, one woman shared that of her 250 Facebook friends, the majority were “people in recovery.” A second woman described a “before-and-after” post where people were sharing side-by-side images of themselves in active addiction versus MOUD treatment. Comments on those images were largely supportive, “like, ‘how amazing, you’re doing so great…you look 10 times better!’” The value of these groups was in feeling less alone—knowing “you’re not the only one [enrolled in MOUD treatment].”

## 4. Discussion

### 4.1. Discussion and Implications for Practice

Our findings on community-level sources of social stigma illustrate moral models of addiction that remain entrenched in society. Women universally shared that a barrier to MOUD engagement was their inability to escape their past identity as substance user, regardless of gains made through treatment. Furthermore, MOUD-based stigma highlights persistent notions of MOUD as a form of active substance use, despite medical guidelines to the contrary [45]. In keeping with the literature, the stigma experienced by women was largely due to gendered associations between OUD, sex work, and parental neglect [23,46]. Women expressed substantial fears associated with consequences of being “outed” as a person in MOUD treatment, and their intent to “pass” as “normal” by maintaining some level of secrecy.

Our findings on the existence of hierarchies within treatment settings and recovery communities that can be discriminatory align with the existing, yet scant, literature [47]. Hierarchies can function as an impediment to peer support and deter women from treatment engagement. Considering the range of hierarchies [48,49] that may be present between women in clinical- and community-based treatment settings may offer important direction for future iterations of women-specific MOUD treatment programming. To be trauma-informed [50], women-specific treatment programs should address relational factors that impact women [51,52,53], and incorporate guidelines for skill-building to improve communication so these groups can be sources of positive social support.

Employment can be an important contributor to sustained recovery and MOUD engagement. Employment is particularly critical for women with OUD, who experience higher rates of socioeconomic insecurity compared to men [53,54,55]. Additionally, employment can be tied to custody requirements for mothers with OUD [56]. Our results indicated workplace environments as sources of discrimination. This finding coincides with emerging studies [57] and suggests a need for anti-stigma interventions and education outreach in the workplace, as well as potential collaborations between treatment and workplace settings (e.g., transportation access and shift flexibility). More research is needed to understand how workplace MOUD-based discrimination may impact men and women differently.

An important discovery was inter-employee workplace stigma in medical settings, which may reinforce fears associated with provider stigma [58] and deter treatment engagement. As such, it may be useful to consider people with OUD as a distinct cultural group, i.e., a group that exposes individuals to different forms of discrimination that, in turn, contributes to health inequities. Following this idea, concepts from cultural humility [59,60] and structural competency [61] may offer strategies to address these barriers to MOUD engagement.

Our findings on social media as a site of community-level discrimination are valuable, given its omnipresence in today’s society. Suicide prevention and other public health efforts have recognized that hopeless media depictions of high-risk individuals may increase suicide attempts [62]. Similarly, condemning people seeking MOUD on social media may deter those considering treatment. Women-specific impacts of social media messaging on MOUD engagement is an underexplored and important avenue of research to explore.

Although social media was identified as a site of discrimination, it was simultaneously a venue where women sought and received encouragement for sustained treatment and recovery. Studies have begun to explore ways people use online platforms for opioid-related help seeking [63,64,65]. Social media and online platforms or forums may have potential to facilitate sustained MOUD engagement. As such, mobile health interventions that leverage these platforms for positive reinforcement and social support may be important community-based complements to clinical treatment protocols. Given increased social isolation, mental health concerns, and substance use risk associated with the COVID-19 pandemic [7] this is a particularly timely avenue to explore. Additionally, health communication efforts that promote “success stories” [29,66] related to long-term treatment engagement could reduce MOUD-related stigma at the community and individual level by normalizing MOUD and promoting positive benefits associated with sustained MOUD engagement.

Women in our study had internalized messaging that MOUD is “substituting one drug for another” or has harmful health impacts. Other studies report how patients perceive methadone to be physically harmful [67,68,69,70,71,72], despite limited or mixed empirical evidence [73,74,75,76,77]. More research is needed to explore these relationships, and to consider the implication of gender on medication side-effects that impact a woman’s physical appearance (e.g., weight gain and tooth decay). Additionally, better understanding synergistic relationships between MOUD and mental health medications is an important line of future research.

In vocalizing fears, perceptions, and experiences related to MOUD pharmacotherapies, women identified a gap in patient–provider education around expectations of treatment duration, and a need for increased shared decision making in regard to MOUD selection. Women also expressed opioid-specific distrust of providers due to prescribing practices and the opioid crisis, and subsequent perceived unethical relationships between providers and the pharmaceutical industry. People that believe in medical conspiracy theories may be less likely to adhere to recommended treatment protocols [78], and is therefore important to examine. Taken together, these findings point to opportunities for improved patient education that could be incorporated into larger health communication interventions.

The extant literature on women and substance use treatment engagement primarily identifies the importance of connecting women to programs that promote safety *external* to treatment settings, such as community-based programs for women experiencing intimate partner violence [18]. Our findings on facilitators of treatment engagement suggest the importance of creating a sense of safety *within* treatment settings. Basic acts of kindness constituted “safety” for women within treatment settings, which is understandable given the vulnerabilities of daily living [18,79] and fractured social relationships experienced by women with active OUD [23,24,25]. Collectively, these findings can guide programmatic interventions and staff trainings to foster a sense of safety within treatment spaces. Lastly, peer workers may be best positioned to address hierarchies within treatment and recovery settings, promote safe treatment environments, and identify relevant women-specific services by offering relatable support [80,81,82]. Increased funding for peer workers and creating opportunities for women to pursue this certification may provide “hope” for those new to treatment [29].

Taken together, women-identified barriers and facilitators to MOUD engagement elicited in this project hold important potential for MOUD engagement and OUD outcomes among this population. Key implications for future research and interventions at the community and clinical level are summarized below (Table 3). Although some of our findings may apply to both men and women, we posit they are experienced differently and require more investigation.

### 4.2. Limitations

Project findings are drawn from a convenience sample of 20 women enrolled in MOUD for at least 90 days. Recruitment was difficult for some group interview sessions, which resulted in occasional “no shows” and smaller groups than planned. Although small, our sample size aligns with norms in qualitative research and provides a depth of data and innovative findings [33,37]. Additionally, research suggests that longer duration of MOUD (e.g., ≥5 years) increases the likelihood of sustained recovery over the subsequent ten years [15]. Because the average length of treatment for our sample was 2.8 years, we did not distinguish factors associated with extended MOUD engagement, highlighting an area for future research.

## 5. Conclusions

We still know relatively little as to why or how women who do access MOUD remain engaged with it over time. Findings presented throughout this article provide critical context on the experiences of women in MOUD treatment, and are important additions to the substance use literature. Novel barriers to treatment engagement identified by women include community-based discrimination as experienced via social media and in the workplace; internalized stigma among MOUD patients that creates hierarchies within treatment settings; opioid-specific mistrust of providers; and women-specific perceptions of MOUD side effects, synergies, and treatment duration. We close by identifying facilitators to treatment engagement, including the importance of cultivating a sense of safety in treatment settings, the value of integrating peer workers into clinical settings, and the potential benefit of social media and other online platforms. In sum, project findings identify key implications for research and interventions to promote MOUD engagement for women with OUD. As treatment access continues to expand in response to COVID-era innovations regarding MOUD, addressing project-identified barriers and facilitators collectively at the community and clinical level hold potential for innovative patient-centered care and increased MOUD engagement for women with OUD.

## Figures and Tables

**Figure 1 ijerph-19-09346-f001:**
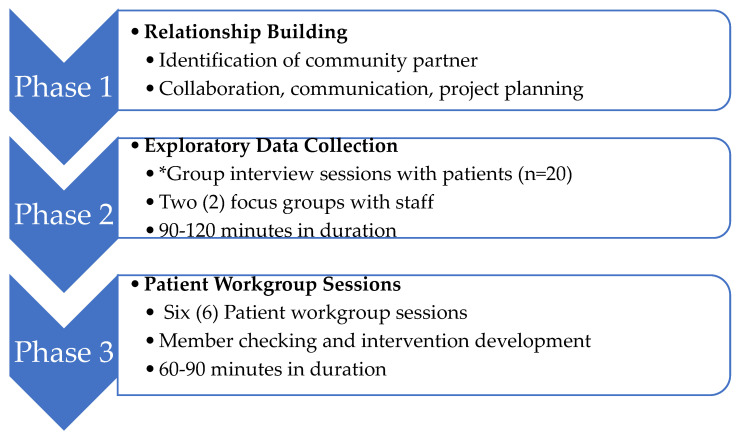
CPPR Project Phases: Adaptation of Digital Storytelling Intervention to Enhance Engagement for Women Enrolled in Opioid Treatment Programs (OTP). * Source of data presented in this manuscript.

**Table 1 ijerph-19-09346-t001:** Participant characteristics.

	Women (*n* = 20)
Age (mean ± SD)	36.6 ± 9.5 years
**Race/ethnicity**	**% (*n*)**
White, Non-Hispanic	65% (13)
Black, Non-Hispanic	5% (1)
Hispanic or Latina/x	30% (6)
**Educational attainment**	**% (*n*)**
<High school/HiSET	5% (1)
High school/HiSET/Vocational	30% (6)
Some college/Associate’s degree	60% (12)
Bachelor’s degree	5% (1)
**Employment status**	**% (*n*)**
Employed full-time	20% (4)
Employed part-time	15% (3)
Laid off/Unemployed	30% (6)
Disabled and not working	25% (5)
Retired and not working	10% (2)
**Household income in last 12 months**	**% (*n*)**
<$10,000	45% (9)
$10,001–$20,000	35% (7)
$20,001–$40,000	15% (3)
$40,001–$75,000	5% (1)
**Opioid use (mean** **± SD)**	
Age at 1st initiation of opioid use	26.6 ± 7.1 years
Average duration of opioid use	4.6 ± 5.0 years
**Current MOUD treatment**	
Age at 1st first treatment (mean ± SD)	31.3 ± 11.1 years
Current MOUD treatment duration (mean ± SD)	2.8 ± 2.4 years

**Table 2 ijerph-19-09346-t002:** Key thematic findings, barriers and facilitators to MOUD engagement.

**Barriers** **1.** **Interpersonal and Community-Based Social Stigma** 1.1 Family and social networks 1.2 Workplace settings 1.3 Social media 1.4 Hierarchies within the treatment and recovery community **2.** **Fear, Perceptions, and Experiences with MOUD Pharmacotherapies** 2.1 Fear of side effects and medication synergies 2.2 Internalized MOUD-Related Stigma 2.3 Expectations of treatment duration 2.4 Opioid-specific provider mistrust
**Facilitators** Sense of safety in treatment settings Support from online communities

**Table 3 ijerph-19-09346-t003:** Implications for Research and Intervention.

**Community level** 1. Health communication efforts that promote treatment “success” stories 2. Mobile health interventions to promote group social support for treatment engagement for women with OUD 3. Outreach and education to address discrimination in workplace and online environments **Clinical level** 4. Patient education around the concept of OUD as a chronic condition, the importance of MOUD for treating OUD, and realistic expectations for treatment duration 5. Shared decision making for MOUD selection between patients and providers 6. Interventions that assess and address social hierarchies within treatment settings 7. Understand MOUD side effects and interactions with mental health pharmacotherapies 8. Integrate peer workers into treatment settings 9. Examine and address opioid-specific provider mistrust

## Data Availability

Not applicable.

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
