# Peer review of "Women-Reported Barriers and Facilitators of Continued Engagement with Medications for Opioid Use Disorder"

_ijerph, 2022, doi:10.3390/ijerph19159346_

Round 1

Reviewer 1 Report

A few spelling/grammar errors: Beginning of the results -- the word "date" is used instead of "data" . Pg. 6, under 3.2.1 "remain engagement" should be "remained engaged". 

Overall, the scientific soundness of this paper would be increased by not using phrases like " many participants " - instead write out how many participants. Also, the quotes are not supported by a deidentified participant number or pseudonym. We wouldn't know if most of this data is coming from one or two sources. 

The results related to self-stigma and social media related stigma are new and a great addition to the literature. The discussion brings out these points well. However, in the results section entitled "Social media" the authors point out that their results are important. This is a statement that should be couched in the discussion and contrasted with existing literature. In general, the opening sentences of the results section make qualitative statements about the results instead of reporting the results - decreasing the scientific nature of the paper

The second sentence of the conclusion is not readable.

Some references are not in proper format.

Author Response

Dear Reviewer One,

We greatly appreciate your valuable feedback, which has helped to substantially strengthen our initial submission. Please see attachment for cover letter and point by point response to your comments.

Reviewer 2 Report

This manuscript entitled “Women-reported barriers and facilitators of continued engagement with medications for opioid use disorder” aimed to identify the barriers and facilitators that adult women with opioid use disorder encountering when engaged in the medication treatment for their OUD. Overall, this is a relevant topic in the field, and it requires more works and efforts in this area to better guide the clinical practice. While the reviewer appreciates the importance of this work, there are several notable concerns in the study design, methods and other areas in the manuscript needed to be clarified in order to ensure the validity of the findings. Areas could be improved to strengthen the manuscript and study are listed below.

1. (Page 1, Introduction, paragraph 1): Reference “Ochlalek et al., 2020” is accidently underlined. Please correct it.  

2. (Page 2, Data collection, paragraph 2): Authors stated that participants completed brief survey on demographics and questionnaires and then received interviews. Please include the survey sample and interview question sample in appendix. Readers may be interested in knowing the type of demographics collected and questions asked in this study.

Have authors assessed the validity and reliability of interview questions? Authors should comment on whether the interview is able to identify the info what it intends to measure and how reliable the results are. This is a key point to mention. If the interview questions are not sufficiently valid or reliable, all the discussions and conclusions in this manuscript are pointless. Please include the relevant info.

3. (Page 3, Results): Authors identified several barriers and facilitators under the results section but didn’t describe the frequency of these barriers and facilitators mentioned by the interviewed women. My understanding is that some barriers and facilitators may be more frequently described than the others and should deserve more attention. Authors should mention the frequencies that each barrier and facilitator was described by the women (e.g., xx women have identified workplace environment as sites of discrimination). With that saying, the hierarchy of barriers and facilitators should be stated.

4. (Page 7, Table 1): Please add “n” to the category variables. Only % is presented in the table now.

5. (Page 7, Table 2): For barrier “1.1 family and social networks” and “2.2 internalized MOUD-Related Stigma”, I didn’t see the discussions about these 2 barriers under results section. Please add them accordingly.

6. (Page 9, Limitations): If authors couldn’t comment on the interview’s validity and reliability, it would be deemed as the biggest limitation of the study. Please add it here.

7. (Page 10, References): Format of some reference such as ref 4, 52, 53, 58, and 74 is not consistent with the others. Please unify them.

Author Response

Dear Reviewer Two,

We greatly appreciate your valuable feedback, which has helped to substantially strengthen our initial submission. Please see attachment for cover letter and point by point response to your comments.
